# INTERACTION-BASED DISENTANGLEMENT OF ENTITIES FOR OBJECT-CENTRIC WORLD MODELS

**Akihiro Nakano, Masahiro Suzuki, and Yutaka Matsuo**
Graduate School of Engineering
The University of Tokyo
7-3-1 Hongo, Bunkyo-ku, Tokyo-to, Japan
{nakano.akihiro,masa,matsuo}@weblab.t.u-tokyo.ac.jp

## ABSTRACT

Perceiving the world compositionally in terms of space and time is essential to understanding object dynamics and solving downstream tasks. Object-centric learning using generative models has improved in its ability to learn distinct representations of individual objects and predict their interactions, and how to utilize the learned representations to solve untrained, downstream tasks is a focal question. However, as models struggle to predict object interactions and track the objects accurately, especially for unseen configurations, using object-centric representations in downstream tasks is still a challenge. This paper proposes STEDIE, a new model that disentangles object representations, based on interactions, into interaction-relevant relational features and interaction-irrelevant global features without supervision. Empirical evaluation shows that the proposed model factorizes global features, unaffected by interactions from relational features that are necessary to predict the outcome of interactions. We also show that STEDIE achieves better performance in planning tasks and understanding causal relationships. In both tasks, our model not only achieves better performance in terms of reconstruction ability but also utilizes the disentangled representations to solve the tasks in a structured manner.

## 1 INTRODUCTION

Humans have evolved to perceive the world in a structured manner, such that we can infer unseen objects, predict their interaction with the environment, and plan to use them to perform certain tasks. Previous works have emphasized achieving similar levels of systematic generalization in deep learning (Goyal & Bengio, 2020) requires a new set of inductive biases that would enable a model to perceive the world as a composition of objects and their relationships (Greff et al., 2020). Recent works on object-centric learning have devised generative models to achieve spatial disentanglement; i.e., these methods decompose images into individual objects in an unsupervised manner (Kosiorek et al., 2018; Greff et al., 2017; Van Steenkiste et al., 2018; Hsieh et al., 2018; Lin et al., 2020; Kossen et al., 2020). This is achieved with the introduction of various inductive biases, such as object propagation and discovery (Kosiorek et al., 2018; Greff et al., 2017; Van Steenkiste et al., 2018), temporal decomposition (Hsieh et al., 2018), background modeling (Lin et al., 2020), and object-interaction modeling (Kossen et al., 2020).

Given the success of object-centric learning, several studies have investigated its effectiveness in learning world models (Ha & Schmidhuber, 2018) to solve downstream tasks (Veerapaneni et al., 2020; Watters et al., 2019a; Kossen et al., 2020; Min et al., 2021). As object-centric learning represents the scene as a composition of objects and its interactions, it would drastically reduce the complexity of modeling the temporal evolution of the scene and, therefore, help in predicting the future and planning. For example, Veerapaneni et al. (2020) proposes OP3, the first fully probabilistic object-centric and action-conditioned video prediction model. They show that it can model interactions between entities and plan interactions to solve simple block-stacking tasks. Nevertheless, prior works have still struggled with solving downstream tasks, as modeling object interactions accurately and keeping track of objects as individuals are common challenges. Also, few studies have proposed models conditioned on actions (Veerapaneni et al., 2020; Kossen et al., 2020).

Table 1: Comparison of relevant work done on object-centric generative models and its utilization for model-based RL. The models can be categorized based on whether it models interactions, is action-conditioned or not, and type of factorization of object representations. OP3 (Veerapaneni et al., 2020) is similar to our work as it models object interactions and actions by an agent, but it represents objects as a single latent variable. PARTS (Zoran et al., 2021) is another work that, like OP3, uses single latents, but the generative process is not conditioned on actions. DDPAE (Hsieh et al., 2018) is also close to our work as it learns factorization implicitly, but interactions are not modeled and is not action-conditioned. For thorough comparison, see Appendix G.

| Method | Models interactions | Action-Conditioned (Plannable) | Type of factorization |
|---|---|---|---|
| ViMON | ✗ | ✗ | Single |
| OP3, PARTS | ✔ | ✔ | Single |
| SQAIR, SCALOR | ✗ | ✗ | Explicitly factorized |
| GSWM | ✔ | ✗ | Explicitly factorized |
| STOVE | ✔ | ✔ | Explicitly factorized |
| DDPAE | ✗ | ✗ | Implicitly factorized |
| **STEDIE (ours)** | ✔ | ✔ | **Implicitly factorized** |

To improve the generation quality of videos and downstream performances, we introduce *interaction-based disentanglement*, which aims to factorize the representations of entities into interaction-relevant relational features and interaction-irrelevant global features. Here, interaction relevance refers to a feature affecting the future properties of other objects via interaction; e.g., the weight of objects is essential to determining whether an object would be moved after contact occurred. At the same time, some objects' features would not be affected via interaction, e.g., the shape of a rigid body would remain unchanged. Importantly, these factorizations must be fully unsupervised to develop a model that can handle various causal relationships, as hand-crafting decomposition for each task is an infeasible burden. Table 1 summarizes relevant earlier research (Hsieh et al., 2018; Veerapaneni et al., 2020; Zoran et al., 2021; Kossen et al., 2020). Although learning single representation or explicit factorizing object representations for model-based RL has been introduced in prior investigations, implicit factorization has not been explored.

To enable interaction-based disentanglement, we propose SpatioTEmporal Disentanglement from Interaction of Entities (STEDIE): a fully-probabilistic, object-centric, and action-conditioned video prediction model. By designing a generative and inference model that implement interaction-based disentanglement, STEDIE disentangles videos both spatially into objects and temporally into relational features and global features. Whereas previous works (Li & Mandt, 2018; Hsieh et al., 2018) have factorized into time-varying and time-invariant features, we model object interactions using neural networks to motivate temporal factorization into *interaction-relevant* and *interaction-irrelevant* features. The model can be trained using only raw input video data in an end-to-end fashion with standard evidence lower bound (ELBO). In our experiments, we first demonstrate the model's ability to disentangle videos spatiotemporally. Furthermore, to verify if decomposing in such a way helps solve downstream tasks, we evaluate the trained model on the planning task and causal relationship understanding task. In the planning task, combined with the cross-entropy method (Rubinstein & Kroese, 2014), we show that STEDIE can perform complex tasks (building block towers) better (up to 13%) and approximately 2x more efficiently than OP3. For causal understanding, we use CausalMBRL (Ke et al., 2021), a recently proposed benchmark of video prediction requiring an understanding of causal relationships. We show that it outperforms various baselines, including a standard VAE-based model, object-centric method trained with pixel-based loss function (CSWM (Kipf et al., 2020)), and OP3 by a large margin.

## 2 RELATED WORKS

There has been many research on using Variational Autoencoders (VAE) (Kingma & Welling., 2013) to learn object-factorized representations for both images and videos with multiple objects. An object-centric generative model trained on videos should learn disentangled representations about not only the objects but also its dynamics, which would lead to better generation of unseen objects or dynamics and understanding of the scenes. Expanding the data domain temporally has been confronted by two major challenges: keeping track of each object and modeling object interactions, such as collision. To overcome these hurdles, there have been mainly three approaches in designing

object representations: (1) learning a single representation per object, (2) learning explicit factorization of representation, and (3) learning factorization of representations unsupervised.

First, many studies have used architectures shown to be effective for static images to extract object-level representations from videos (Weis et al., 2020; Veerapaneni et al., 2020; Watters et al., 2019a; Creswell et al., 2020; Min et al., 2021; Zoran et al., 2021). For example, ViMON (Weis et al., 2020) expands MONet (Burgess et al., 2019) to videos by adding a gated recurrent unit after the encoder to aggregate information over time for each object. OP3 (Veerapaneni et al., 2020) uses IODINE as the inference model, with additional modules to capture interactions between objects and action interventions.

Secondly, another branch of research has derived from AIR (Eslami et al., 2016), a model that learns several predefined factors of objects ("what", "where", and "presence") for a static image to obtain further disentangled object representations (Kosiorek et al., 2018; Hsieh et al., 2018; Jiang et al., 2019; Kossen et al., 2020; Crawford & Pineau, 2020; Lin et al., 2020). These models, similar to AIR, learn a set of latent variables per object, designed by humans before training, for better object tracking. Introduction of "velocity" and physical interaction (Kossen et al., 2020) or "layer" and background (Jiang et al., 2019; Lin et al., 2020) has improved AIR-derived models' ability to keep track of objects even in cases of occlusion or interaction.

Thirdly, unsupervised time-wise disentanglement of videos into content and pose of objects has been studied (Li & Mandt, 2018; Hsieh et al., 2018; Zablotskaia et al., 2021; Li et al., 2021). Using VAEs, Hsieh et al. (2018) proposed learning a content vector and a low-dimensional vector to predict dynamics for each object in a video, but do not model interactions between objects. The work by Li & Mandt (2018) is closest to ours; it proposes a model that learns to disentangle the scene temporally using bi-LSTM (Graves & Schmidhuber, 2005) architecture, but it is limited to a single object per scene. Temporal disentanglement into time-varying and time-invariant features has been studied in various domains as well, including texts into topics and details (Bowman et al., 2015), speeches into speaker and linguistic features (Hsu et al., 2017), and images into content and detailed textures (Chen et al., 2016). Unsupervised disentanglement of object representations is also related to how humans recognize the "sameness" of an object (Baillargeon, 1987; Spelke, 1990; Aguiar & Baillargeon, 1999; Spelke et al., 1992; Hespos & Baillargeon, 2001; Spelke & Kinzler, 2007; Spelke, 2013).

Among these three categories, our proposal falls in the third category. Besides, we aim to learn object-centric world models (action-conditioned video prediction models), and to develop a model that can be used on downstream tasks, such as multi-step planning. As explained in Table 1, while single-representation modeling and explicit-factorization modeling methods have been extended for usage in model-based RL (Veerapaneni et al., 2020; Watters et al., 2019a; Kossen et al., 2020; Min et al., 2021; Zadaianchuk et al., 2021; 2022), using unsupervised-factorization modeling methods has not been explored. In this paper, we reconsider and broaden temporal disentanglement to object interactions as the decomposition into time-invariant, interaction-irrelevant features and time-dependent, interaction-relevant features.

Apart from generative models, models trained using expectation maximization (Greff et al., 2016; 2017; Van Steenkiste et al., 2018), pixel-based losses (Battaglia et al., 2016; Watters et al., 2017; Xu et al., 2019; Piloto et al., 2022), and contrastive learning (Kipf et al., 2020) have been proposed. More recently, slot attention (Locatello et al., 2020) has been used to extract object-centric features and predict its dynamics (Kabra et al., 2021; Kipf et al., 2021; Elsayed et al., 2022). For temporal disentanglement, approaches using generative adversarial networks (Goodfellow et al., 2020) and autoencoders have been studied (Tulyakov et al., 2018; Villegas et al., 2017; Denton et al., 2017). As we are interested in learning disentangled representations that can be utilized for downstream tasks, we do not consider competing reconstruction ability with these models as the main objective of the present work.

## 3 SPATIOTEMPORAL DISENTANGLEMENT FROM INTERACTIONS OF ENTITIES

Our goal is to build a probabilistic model for a sequence of images, $x^{(0:T)}$, using Markovian latent variables for each timestep, which are further decomposed into object-centric representations for $K$ distinct objects. As we are interested in utilizing the trained model for model-based RL to solve downstream tasks, we assume an invisible agent with a fixed point of view that can interact with the environment. The actions taken by the agent is expressed as $a^{(0:T-1)}$; each action consists of moving an object at a certain position to another coordinate. In this paper, we refer the object itself in the real world as *objects* and its latent representations acquired through the model as *entities*.

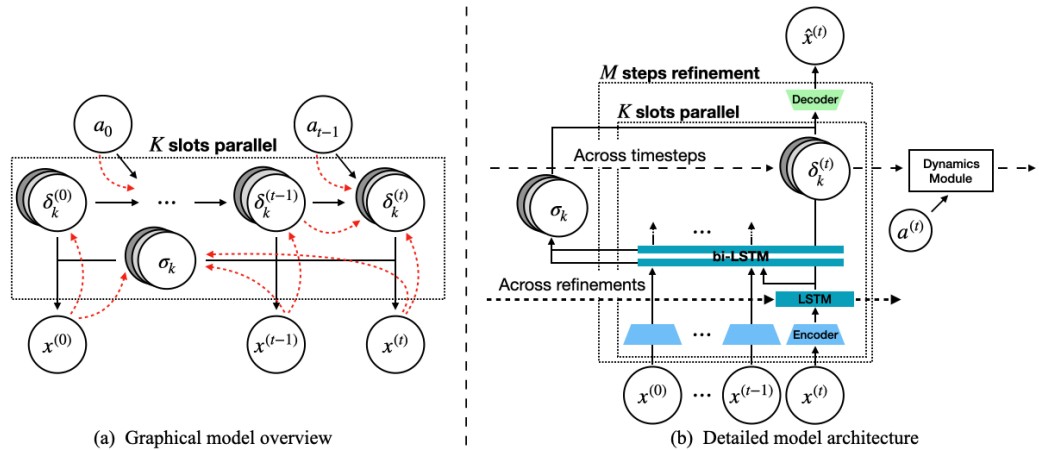

(a) Graphical model overview                       (b) Detailed model architecture

Figure 1: $\sigma_{1:K}$ and $\delta_{1:K}^{(t)}$ denote global features and relational features respectively. (a) Graphical model visualization of STEDIE. Solid black arrow indicates generation process and dashed red arrow indicates inference procedure. (b) Model architecture of STEDIE. In the inference process, images and actions up to the timestep of interest are used as inputs. To generate at a certain timestep, the relational feature at that timestep is concatenated with the global feature. As two features are inferred independently, the global feature inferred at a previous timestep can be reused.

Figure 1 (a) shows the graphical model of our proposed model, STEDIE. STEDIE learns to disentangle entities into interaction-irrelevant (*global* features, $\sigma_{1:K}$) and interaction-relevant features (*relational* features, $\delta_{1:K}^{(t)}$). We hypothesize that entities can be factorized based on interactions as object dynamics can be predicted from the object's partial information. Other features should be learned as they operate as "keys" to identify the same object across time. It is essential to learn to model this factorization implicitly to reduce assumption of the data. For example, the assumption that object size remains unchanged is valid for 2D data but fails for 3D data as objects can move towards or away from the observing agent.

As our model builds on OP3 (Veerapaneni et al., 2020), we first explain about OP3 and then proceed to explain the extension we incorporated to learn global and relational features for each entity.

### 3.1 PRELIMINARY: OBJECT-CENTRIC PERCEPTION, PREDICTION, AND PLANNING (OP3)

OP3—Object-centric Perception, Prediction, and Planning—is a generative model that can decompose videos $x^{(0:T)}$ into $K$ objects $z_{1:K}^{(0:T)}$, predict the effect of actions $a^{(0:T-1)}$ by an agent and object interactions, predict future states, and use obtained representations to solve downstream planning tasks (Veerapaneni et al., 2020). The model consists of an observation module, $p_\theta(x^{(0:T)}|z_{1:K}^{(0:T)})$, and a dynamics module, $p_\theta(z_{1:K}^{(t)}|z_{1:K}^{(t-1)}, a^{(t-1)})$, which are trained using variational inference and an inference model, $q_\phi(z_{1:K}^{(0:T)}|x^{(0:T)}, a^{(0:T)})$. IODINE (Greff et al., 2019) architecture is used as the inference model. The dynamics module is reused for inference to reduce the number of parameters and training time. For better object-wise decomposition, OP3 proposes an interactive inference algorithm, which updates the posterior and the belief over time. OP3 claims that interactive inference encourages the model to disambiguate objects and ensure temporal consistency. Overall, the generative model and inference model can be expressed as $p_\theta(x^{(0:T)}, z_{1:K}^{(0:T)}|a^{(0:T-1)}) = p(z_{1:K}^{(0)}) \prod_{t=1}^{T} p_\theta(z_{1:K}^{(t)}|z_{1:K}^{(t-1)}, a^{(t-1)}) \prod_{t=0}^{T} p(x^{(t)}|z_{1:K}^{(t)})$ and $q_\phi(z_{1:K}^{(0:T)}|x^{(0:T)}, a^{(0:T-1)}) = \prod_{t=0}^{T} q_\phi(z_{1:K}^{(t)}|z_{1:K}^{(t-1)}, x^{(t)}, a^{(t)})$, respectively. Although iterative inference of IODINE and the proposed interactive inference motivates the model to recognize and keep track of objects, experimental results (Appendix E) show failure cases in which objects move around slots or its color changes over time. We think this is due to dynamics module using full representation of objects; small variances are accumulated over time to cause changes to object features that should remain unaffected by the interactions.

### 3.2 STEDIE

**Generative Model.** The generation process of our model is included in Figure 1 (b). STEDIE assumes that an image at timestep $t$, $x^{(t)}$, is generated from relational features, $\delta_{1:K}^{(t)}$ and global

features, $\sigma_{1:K}$ of $K$ distinct objects. Note that because global features are unaffected by interactions, it remains the same regardless of timestep. Hence, the generative model of observing $T$ frames of scenes $x^{(0:T)}$ and latent entities $\delta_{1:K}^{(0:T)}, \sigma_{1:K}$ from taking actions $a^{(0:T-1)}$ consists of (1) observation module $p_\theta(x^{(t)}|\delta_{1:K}^{(t)}, \sigma_{1:K})$ that renders the image at timestep $t$ conditioned on relational features features $\delta_{1:K}^{(t)}$ and global features $\sigma_{1:K}$ and (2) dynamics module $p_\theta(\delta_{1:K}^{(t)}|\delta_{1:K}^{(t-1)}, a^{(t-1)})$ that models object interactions in latent space. Overall, the generative model is expressed as,

$$
\begin{aligned}
&p_\theta(x^{(0:T)}, \delta_{1:K}^{(0:T)}, \sigma_{1:K}|a^{(0:T-1)}) \\
&= p_\theta(\sigma_{1:K})p_\theta(\delta_{1:K}^{(0)}|\sigma_{1:K}) \prod_{t=1}^{T} p_\theta(\delta_{1:K}^{(t)}|\delta_{1:K}^{(t-1)}, a^{(t-1)}) \prod_{t=0}^{T} p_\theta(x^{(t)}|\delta_{1:K}^{(t)}, \sigma_{1:K}),
\end{aligned}
\tag{1}
$$

where $p(\sigma_{1:K}^{(0)})$ is a Gaussian distribution with mean $\mathbf{0}$ and variance $\mathbf{I}$, and $p_\theta(\delta_{1:K}^{(0)}|\sigma_{1:K})$ is modeled with an MLP. We condition relational features on global features as these two features are not completely independent. In most cases, static feature affects how the objects move, as the 2D weighted-block pushing dataset (Ke et al., 2021) we use in our experiments is an explicit example. We model the observation module using a BroadcastCNN (Watters et al., 2019b). Each entity representations $\delta_k^{(t)}, \sigma_k$ are decoded into pixel-wise mean, $\boldsymbol{\mu}_k$, and segmentation mask, $\boldsymbol{m}_k = p(C = k|\delta_k^{(t)}, \sigma_k)$, to model the likelihood, $p_\theta(x^{(t)}|\delta_{1:K}^{(t)}, \sigma_{1:K}) = \sum_{k=1}^{K} \boldsymbol{m}_k \cdot \mathcal{N}(x^{(t)}; \boldsymbol{\mu}_k, \boldsymbol{\Sigma}^2)$, with the same variance $\boldsymbol{\Sigma}^2$ for all pixels. Following OP3, we approximate the effect of the intervention on the entities by an action $a^{(t)}$, $p_\theta(\delta_{1:K}^{(t)}|\delta_{1:K}^{(t-1)}, a^{(t-1)})$, by breaking into pairwise interactions. The entire dynamics module is modeled as, using MLP-based function, $\ell$, $p_\theta(\delta_k^{(t)}|\delta_k^{(t-1)}, a^{(t-1)}) = \ell(\delta_k^{(t)}|\delta_k^{(t-1)}, a^{(t-1)}, \delta_k^{interact})$ where $\delta_k^{interact} = \sum_{i \neq k}^{K} d_{oo}(\delta_i, \delta_k)$. We apply the same function, $d_{oo}$, for each $k = 1, \cdots, K$ to calculate pairwise entity interactions. While full latent variables were used to predict the dynamics in previous studies, our model only takes the relational features, $\delta_{1:K}^{(t)}$. This motivates the model to learn object features unaffected by interactions as global features.

**Inference Model and Learning.** The inference process of our model is also included in Figure 1 (b). The inference model is,

$$
\begin{aligned}
&q_\phi(\delta_{1:K}^{(0:T)}, \sigma_{1:K}|x^{(0:T)}, a^{(0:T)}) \\
&= q_\phi(\sigma_{1:K}|x^{(0:T)}, a^{(0:T-1)})q_\phi(\delta_{1:K}^{(0)}|x^{(0)}) \prod_{t=1}^{T} q_\phi(\delta_{1:K}^{(t)}|\delta_{1:K}^{(t-1)}, x^{(t)}, a^{(t-1)}),
\end{aligned}
\tag{2}
$$

which is factorized into two parts, (1) inferring the global features from the entire sequence, $x^{(0:T)}, a^{(0:T)}$, and (2) inferring the relational features per timestep. As a single inference to infer the parameters at a certain timestep is insufficient to break the symmetry to decompose a scene into entity representations (Kipf et al., 2020), we incorporate iterative inference in several aspects. Following OP3, we use iterative inference (Marino et al., 2018) to refine the parameters at each timestep using IODINE (Greff et al., 2019). Iterative inference updates the posterior parameters using gradient information about the ELBO obtained by the current estimate of the parameters. The gradients pass information of what other slots have not learned, therefore encouraging slots to capture different pixels of the image. We also incorporate interactive inference as OP3 and update the prior and posterior at each timestep. This ensures keeping track of the "same" object using interactive inference for both global and relational features (Appendix F).

However, this causes difference in the expressive power of global and relational features as all timesteps up to the current timestep are used to infer global features while relational features are inferred from the current timestep only. Hence, to compensate this problem, we infer global features only at the first refinement step at each timestep. Note that we must infer at the first step, as IODINE requires gradient information about the ELBO. To infer the global variable, we process the outputs of IODINE through a bi-LSTM network (Graves & Schmidhuber, 2005). To infer object dynamics, we reuse $p_\theta(\delta_{1:K}^{(t)}|\delta_{1:K}^{(t-1)}, a^{(t-1)})$ to reduce the number of parameters and training time (Kossen et al., 2020; Veerapaneni et al., 2020).

As the inference of global and relational features are independent at each timestep, it is also possible to update only either of the features. Inferring features independently allows efficient use of the

model during test-time; we infer both features initially, update only the relational features to predict interactions, and then combine the features to reconstruct a scene at any timestep.

Finally, parameters of a generative model, $p_\theta$, and an inference model, $q_\phi$, are optimized via variational inference to maximize the evidence lower bound (ELBO):

$$
\mathbb{E}_{\delta_{1:K}^{(0:T)},\sigma_{1:K}\sim q_\phi(\delta_{1:K}^{(0:T)},\sigma_{1:K}|x^{(0:T)},a^{(0:T-1)})}\left[\log\frac{p_\theta(x^{(0:T)},\delta_{1:K}^{(0:T)},\sigma_{1:K}|a^{(0:T-1)})}{q_\phi(\delta_{1:K}^{(0:T)},\sigma_{1:K}|x^{(0:T)},a^{(0:T)})}\right]
$$

$$
=\sum_{t=0}^{T}\left(\mathcal{L}_r^{(t)}-\mathcal{L}_c^{(t)}\right)-D_{\mathrm{KL}}(q_\phi(\sigma_{1:K}|x^{(0:T)},a^{(0:T-1)})\|p_\theta(\sigma_{1:K})),
$$

(3)

where $\mathcal{L}_r^{(t)} = \mathbb{E}_{\delta_{1:K}^{(t)},\sigma_{1:K}\sim q_\phi(\delta_{1:K}^{(t)},\sigma_{1:K}|\delta_{1:K}^{(0:t-1)},x^{(0:t)},a^{(0:t-1)})}[\log p_\theta(x^{(t)}|\delta_{1:K}^{(t)},\sigma_{1:K})]$ and $\mathcal{L}_c^{(t)} = D_{\mathrm{KL}}(q_\phi(\delta_{1:K}^{(t)}|\delta_{1:K}^{(t-1)},x^{(t)},a^{(t-1)})\|p_\theta(\delta_{1:K}^{(t)}|\delta_{1:K}^{(t-1)},a^{(t-1)}))$. See appendix for derivation of the loss function (Appendix A) and details on model architecture (Appendix B).

## 4 EXPERIMENTS

We evaluate STEDIE using two datasets each with different characteristics: (1) 3D block moving dataset (Blocks) from Janner et al. (2019) and Veerapaneni et al. (2020), and (2) 2D weighted-block pushing dataset (Weights) from Ke et al. (2021). Using these datasets, we conduct three experiments aimed to answer the following questions, (1) Have the global and relational features learned different information? and (2) Does modeling to learn disentangled features lead to better performance in downstream tasks, such as planning and causal understanding?

To answer the first question, we conducted latent swapping experiment (subsection 4.1). Specifically, given two unseen samples, we swapped each slot's relational feature accordingly and reconstructed the samples. Since all objects are identical except for their color, we expect features such as position to be learned as relational features and features such as color to be learned as global features. If the model has learned to disentangle, generating with the swapped relational features will result in sequences with the same object colors but different positions.

We then analyzed the impact of learning disentangled representations on two downstream tasks (subsection 4.2). First, we evaluated the learned representations on planning. We used the block stacking task included in Blocks dataset, and used the models to plan the stacking process combined with model-predictive control (MPC) (Finn & Levine, 2017) planning. Intuitively, separating interaction-irrelevant information should help the model plan better, because objects follow the same dynamics regardless of their interaction-irrelevant characteristics. Secondly, we used the Weights dataset to evaluate whether learning global and relational features contribute to better understanding of underlying causal relationships between objects in a scene. In this experiment, we are interested in investigating whether interaction-based disentanglement is effective for a more complex environment. The trained model should capture object color as an interaction-relevant feature to utilize to learn causal structure present in the data. We compared STEDIE with previous works in this dataset only, as AIR-based models (ex. STOVE) are known to struggle with data that include background information (Lin et al., 2020). See appendix for further details on training setup, datasets, and downstream tasks (Appendix C).

### 4.1 SWAPPING LATENT VARIABLES

Figure 2 visualizes an example of generated sequences with originally inferred and swapped global features. By reconstructing the sequence using the swapped global features, while retaining the relational features results, the color of the objects were exchanged, but their positions remained the same. This suggests that STEDIE successfully disentangled representations into relational features that change through interactions, such as position, and global features that remain unchanged throughout interactions, such as color. We also observed that, due to the disentanglement, STEDIE was able to keep track of objects better even when rolling out longer timesteps (Appendix E).

We further evaluated quantitatively how much object color and position match after swapping the global and relational variables respectively (Table 2). We calculated mean squared error (MSE) of average RGB color and intersection over union (IoU) of the slots' masks be-

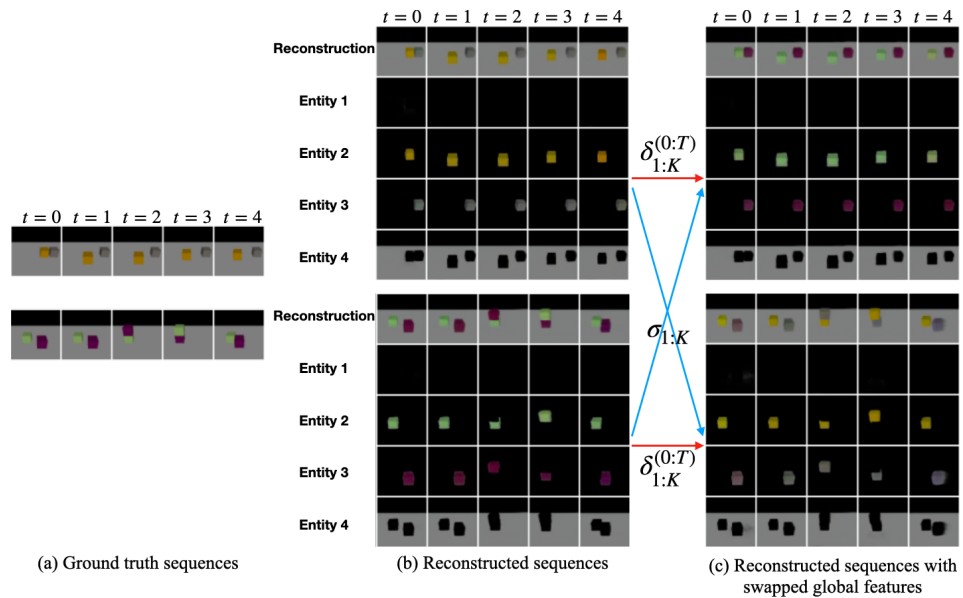

(a) Ground truth sequences  (b) Reconstructed sequences  (c) Reconstructed sequences with swapped global features

Figure 2: Visualization of swapping global features between two sequences. The columns represent timesteps. Blue arrows indicate swapped global features and red arrows indicate unswapped relational features. In (b) and (c), top row represents the reconstructed whole image and the second to fifth rows represent the the reconstructed image for each slot. Slots are sorted in ascending order of mask size and global features are swapped between the slots with the same slot #.

tween the reconstructed sequences before and after swapping the latent variables. As the objects are the elements that differ between sequences, we calculate the metrics for only the slots which contain the object representations by using mask size as threshold. As per Table 2, color MSE of the original sequence and the sequence with the global variable swapped is low, whereas having the same relational variable resultss in higher IoU.

Overall, these results show that STEDIE learns to disentangle sequences both spatially into object representations and temporally into time-varying and time-invariant features. The experiment showed that object colors are preserved between sequences with same global variable and object configurations are preserved between sequences with same relational variable. Since entities are learned as two different latent variables, the model is capable of freely combining variables between sequences, suggesting how STEDIE captures the scene in a structured manner.

Table 2: Color MSE and mask IoU on swapping global or relational latent variables between two sequences in the test dataset using STEDIE. Lower MSE indicate that color are similar before and after swapping the feature. Higher IoU indicates a similar position.

| Swapping feature | MSE ($\downarrow$) | IoU ($\uparrow$) |
|---|---|---|
| Swap relational | **0.0838** | 47.4 |
| Swap global | 0.1574 | **62.8** |

## 4.2 EFFECT ON DOWNSTREAM TASKS

**Planning.** We compare planning results with OP3 (Veerapaneni et al., 2020). Given representations of current timestep, $h_{1:K}^{(t)} = Concat(\sigma_{1:K}, \delta_{1:K}^{(t)})$, and inferred goal representations, $h_{1:K}^g$, a cost function can be defined as, $\mathcal{C}(h_{1:K}^{(t)}, h_{1:K}^g) = \sum_{k \in \mathcal{S}^{(t)}} \min_{l \in \mathcal{S}^g} c(h_k^{(t)}, h_l^g)$, where $\mathcal{S}^{(t)}$ and $\mathcal{S}^g$ denote predicted entities of current timestep and goal image, respectively and $c(\cdot)$ is some distance function. Assuming that a single step is enough to move a certain object to its goal position, we can greedily choose the actions that achieve $\min_{k \in \mathcal{S}^{(t)}, l \in \mathcal{S}^g} c(h_k^{(t)}, h_l^g)$. The actions are sampled and optimized using the cross-entropy method (CEM) (Rubinstein & Kroese, 2014). We evaluated the model using (1) accuracy of $\frac{\text{\# of blocks in the correct position}}{\text{\# goal blocks}}$, where a correct position is based on a threshold of the distance error (Janner et al., 2019), (2) MSE of the reconstructed planned image, and (3) average CEM steps taken when sampling actions. The model was trained using sequences that contained up to two objects.

Table 3: Accuracy (%), MSE, and average CEM steps until reaching the goal state of multi-step planning for building block towers by OP3 and our STEDIE model.

| # Blocks | Model | Acc. ($\uparrow$) | MSE ($\downarrow$) | Average CEM steps ($\downarrow$) |
|---|---|---|---|---|
| 1 | OP3 | **85%** | 0.0014 | 1.17 |
| | STEDIE (ours) | 84% | **0.0012** | **1.00** |
| 2 | OP3 | 58% | 0.0049 | 1.63 |
| | STEDIE (ours) | **63%** | **0.0041** | **1.45** |
| 3 | OP3 | 42% | 0.0102 | 3.50 |
| | STEDIE (ours) | **55%** | **0.0075** | **1.82** |

Table 4: Rollout MSE for the Weights dataset calculated at $t = 1, 5, 10$. Except for OP3 and STEDIE, the models were trained in two stages, in which the encoder and the decoder were trained first without transition and the transition module was trained afterwards. For STOVE, we report the MSE at $t = 5, 10$ only as the latents representing object velocities are estimated by conditioning on the first two timesteps.

| Dataset | Model | Loss | MSE ($\downarrow$) | | |
|---|---|---|---|---|---|
| | | | ($t = 1$) | ($t = 5$) | ($t = 10$) |
| Observed | VAE | ELBO | 167.75 | 329.83 | 365.01 |
| | CSWM | Pixel MSE | 121.91 | 282.18 | 307.91 |
| | OP3 | ELBO | **20.80** | 275.44 | 444.14 |
| | STOVE | ELBO | N/A | 301.49 | 448.93 |
| | STEDIE (ours) | ELBO | 68.44 | **110.76** | **225.30** |
| FixedUnobserved | VAE | ELBO | 111.37 | 269.79 | 310.84 |
| | CSWM | Pixel MSE | 95.57 | 221.61 | 234.32 |
| | OP3 | ELBO | 270.56 | 267.61 | 267.34 |
| | STOVE | ELBO | N/A | 362.23 | 496.70 |
| | STEDIE (ours) | ELBO | **81.51** | **102.18** | **181.05** |
| Unobserved | VAE | ELBO | 110.11 | 263.43 | 318.17 |
| | CSWM | Pixel MSE | 92.57 | 215.49 | 229.85 |
| | OP3 | ELBO | 59.08 | 340.00 | 616.76 |
| | STOVE | ELBO | N/A | 218.10 | 313.22 |
| | STEDIE (ours) | ELBO | **39.68** | **47.94** | **93.18** |

Figure 3 shows some examples of planning results, and Table 3 shows the quantitative results. As the results show, our model exceeds OP3 except for accuracy for when the number of goal blocks is 1. STEDIE achieves higher or equivalent accuracy even for unseen numbers of objects, which suggests that disentangling interaction-relevant and interaction-irrelevant features reduces redundancy when predicting object dynamics, enabling the model to accurately pick and place the objects. Our model also results in lower reconstruction error, indicating that the model "imagines" more accurately how the objects' layout will change given an action. Fewer CEM steps until completion indicate that our model is able to pick the correct actions efficiently. The results especially show improvement in lowering average CEM steps for unseen object configuration (# Blocks=1,3). This supports our hypothesis that separating object representations into global and relational features enhances the model to solve tasks efficiently. By separating irrelevant features, STEDIE finds optimal actions efficiently by predicting interactions and tracking objects more accurately.

Figure 3: Visualization of start, goal, and reconstruction of reached state by OP3 and STEDIE. STEDIE reaches the goal state more accurately.

**Understanding Causal Relationship.** Table 4 summarizes the rollout performance compared to other methods. We compared STEDIE against OP3 and STOVE as well as two baseline models provided by Kipf et al. (2020) with action vectors incorporated in the dynamics module, using reconstruction MSE as the evaluation metric. For OP3, STOVE, and STEDIE, the models were trained to predict up to $t = 5$ timesteps. STOVE was trained by modeling the scene as a composition of $N = 3$ objects. We also used action vectors to condition the dynamics model without training the reward network. For other baseline

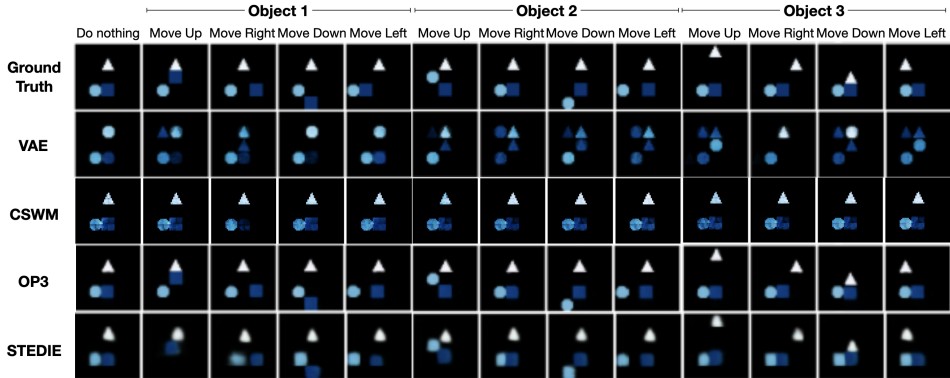

Figure 4: Reconstruction example of every possible single rollout on the Weights dataset (Observed). STOVE result is not included as it only uses the dynamics module starting from $t = 1$. OP3 and our model achieves realistic reconstruction for most cases.

models, the same dataset was broken down into five different single-step rollouts. As Table 4 shows, our method achieved the lowest MSE for most cases, outperforming baseline models which were trained using reconstruction-based loss. STOVE, albeit its explicit factorization of entities, performed worse than STEDIE. This is because STOVE only infers object positions, velocities, and sizes from images $x^{(0:T)}$ while their colors are not explicitly modeled, leading to its inability to understand the causal relationship of objects accurately. Although OP3 achieves the low MSE at $t = 1$ except for FixedUnobserved case, the error increases largely at $t = 5, 10$. This suggests that OP3 has learned to infer a certain timestep accurately using iterative inference, but did not capture the causal relationship and its affect on the dynamics. On the other hand, STEDIE's MSE is consistently lower than the other methods when rolled out further into the future. This result indicates that STEDIE is able to reconstruct well by capturing the causal relationship of objects.

Figure 4 shows the outcome of a single rollout of each object into all possible directions. The figure shows the ability of OP3 and STEDIE to understand causal relationships. Here, the model needs to comprehend the relationship for pushing (1) the dark blue square (object 1) towards the left or (2) the light blue circle (object 2) towards the right. The figure shows that both OP3 and STEDIE have captured this causal relationship. Comparing OP3 and STEDIE, Table 4 and Figure 6 show that STEDIE understands the causal relationship better as it tracks object shape and color throughout the sequence and therefore better overall reconstruction. However, its reconstruction can be improved in the future, as it reconstructs object 2 as a square instead of a circle in the example above.

## 5 CONCLUSION

In this paper, we proposed STEDIE, a model that learns to disentangle an image-action sequence spatially into each object and temporally into interaction-irrelevant global features and interaction-relevant relational features, which was trained using only raw input data without any supervision (Equation 3). The first experiment, latent swapping experiment, showed our model's ability to disentangle object representations in a meaningful way. From purely input images and actions, the model decomposed entities into features based on interactions. The next two experiments confirmed that decomposing scenes spatiotemporally leads to better performance in solving downstream tasks. In the planning task, separating irrelevant information enabled the model to use only positional information to predict interactions. In causal understanding task, the results showed that STEDIE learns the causal relationship between the objects better than other models. OP3 did not factorize entities and STOVE factorized them explicitly, but STEDIE's implicit factorization is a feasible implementation that can be applied to various datasets, through which it achieves a better understanding of object interactions. However, as AIR-based STOVE and spatial-mixture based STEDIE's differ in terms of architecture, implementing explicit factorization using spatial-mixture based models should be explored. Although we experimented on two datasets with different characteristics, further work on designing a more natural and complex dataset should be considered to explore the effects of learning multiple object features without supervision. We also leave extending this work to other architectures for future exploration.

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

## A DERIVATION OF LOSS FUNCTION

Here we provide derivation of the ELBO. As we define generative model and inference model as Equation 1 and Equation 2 respectively, the ELBO (Equation 3) can be factorized across time,

$$
\begin{aligned}
&\mathbb{E}_{\delta_{1:K}^{(0:T)},\sigma_{1:K}\sim q_\phi(\delta_{1:K}^{(0:T)},\sigma_{1:K}|x^{(0:T)},a^{(0:T-1)})}\left[\log\frac{p_\theta(x^{(0:T)},\delta_{1:K}^{(0:T)},\sigma_{1:K}|a^{(0:T-1)})}{q_\phi(\delta_{1:K}^{(0:T)},\sigma_{1:K}|x^{(0:T)},a^{(0:T)})}\right] \\
&= \mathbb{E}_{\delta_{1:K}^{(0)},\sigma_{1:K}\sim q_\phi(\delta_{1:K}^{(0)},\sigma_{1:K}|x^{(0)})}[\log p_\theta(x^{(0)}|\delta_{1:K}^{(0)},\sigma_{1:K})] - D_{\mathrm{KL}}(q_\phi(\delta_{1:K}^{(0)}|x^{(0)})\|p_\theta(\delta_{1:K}^{(0)}|\sigma_{1:K})) + \cdots \\
&\quad + \mathbb{E}_{\delta_{1:K}^{(t)},\sigma_{1:K}\sim q_\phi(\delta_{1:K}^{(t)},\sigma_{1:K}|\delta_{1:K}^{(0:t-1)},x^{(0:t)},a^{(0:t-1)})}[\log p_\theta(x^{(t)}|\delta_{1:K}^{(t)},\sigma_{1:K})] \\
&\quad - D_{\mathrm{KL}}(q_\phi(\delta_{1:K}^{(t)}|\delta_{1:K}^{(t-1)},x^{(t)},a^{(t-1)})\|p_\theta(\delta_{1:K}^{(t)}|\delta_{1:K}^{(t)},a^{(t-1)})) \\
&\quad - D_{\mathrm{KL}}(q_\phi(\sigma_{1:K}|x^{(0:T)},a^{(0:T-1)})\|p_\theta(\sigma_{1:K})) \\
&= \sum_{t=0}^{T}\left(\mathcal{L}_r^{(t)}-\mathcal{L}_c^{(t)}\right) - D_{\mathrm{KL}}(q_\phi(\sigma_{1:K}|x^{(0:T)},a^{(0:T-1)})\|p_\theta(\sigma_{1:K}))
\end{aligned}
$$

$$(4)$$

where

$$
\begin{aligned}
\mathcal{L}_r^{(t)} &= \mathbb{E}_{\delta_{1:K}^{(t)},\sigma_{1:K}\sim q_\phi(\delta_{1:K}^{(t)},\sigma_{1:K}|\delta_{1:K}^{(0:t-1)},x^{(0:t)},a^{(0:t-1)})}[\log p_\theta(x^{(t)}|\delta_{1:K}^{(t)},\sigma_{1:K})] \\
\mathcal{L}_c^{(t)} &= D_{\mathrm{KL}}(q_\phi(\delta_{1:K}^{(t)}|\delta_{1:K}^{(t-1)},x^{(t)},a^{(t-1)})\|p_\theta(\delta_{1:K}^{(t)}|\delta_{1:K}^{(t-1)},a^{(t-1)}))
\end{aligned}
$$

$$(5)$$

## B DETAILED ARCHITECTURE

We use a similar architecture as OP3 (Veerapaneni et al., 2020), with learning global and relational features for each object as the key difference. In this section, we provide details on the model architecture.

Following IODINE (Greff et al., 2019) and OP3, we use the following inputs to IODINE, where LN means Layernorm and SG means stop gradients. The following image-sized inputs are concatenated and fed to the corresponding convolutional network:

| Description | Formula | LN | SG | Ch. |
|---|---|---|---|---|
| image | $\boldsymbol{x}$ | | | 3 |
| means | $\boldsymbol{\mu}$ | | | 3 |
| mask | $\boldsymbol{m}_k$ | | | 1 |
| mask-logits | $\hat{\boldsymbol{m}}_k$ | | | 1 |
| mask posterior | $p(\boldsymbol{m}_k|\boldsymbol{x},\boldsymbol{\mu})$ | | | 1 |
| gradient of means | $\nabla_k\mathcal{L}$ | ✓ | ✓ | 3 |
| gradient of mask | $\nabla_{\boldsymbol{m}_k}\mathcal{L}$ | ✓ | ✓ | 1 |
| pixelwise likelihood | $p(\boldsymbol{x}|\boldsymbol{h})$ | ✓ | ✓ | 1 |
| leave-one-out likelih. | $p(\boldsymbol{x}|\boldsymbol{h}_{i\neq k})$ | ✓ | ✓ | 1 |
| coordinate channels | | | | 2 |
| | | | total: | 17 |

We concatenate the posterior parameters, $\lambda_{1:K}\in\{\delta_{1:K},\sigma_{1:K}\}$, and their gradients with the output of the refinement networks's convolutional part and use the result as input to the LSTM network inside IODINE:

| Description | Formula | LN | SG |
|---|---|---|---|
| gradient of posterior | $\nabla_{\lambda_k}\mathcal{L}$ | ✓ | ✓ |
| posterior | $\lambda_k$ | | |

Below, we denote $R_d, d_a, d_\sigma$, and $d_\delta$ as dimensions of deterministic components of latent variables, action vectors, global features, and relational features, respectively. The architecture of the inference model is a combination of IODINE and a bi-LSTM network as follows:

| Type | Size/Ch. | Act. Fn. | Comments |
|---|---|---|---|
| MLP | $d_\delta$ | | for relational |
| MLP | $d_\sigma$ | Leaky ReLU | for global |
| biLSTM | $64 + d_a$ | Tanh | +actions, for global |
| MLP | 128 | Linear | |
| LSTM | 128 | Tanh | |
| Concat $[\lambda_i, \nabla_{\lambda_i}]$ | $2R_s$ | | |
| MLP | 128 | ELU | |
| Avg. Pool | $R_s$ | ELU | |
| Conv $3 \times 3$ | $R_s$ | ELU | |
| Conv $3 \times 3$ | 32 | ELU | |
| Conv $3 \times 3$ | 32 | ELU | |
| Inputs | 17 | | |

Latent variables are concatenated along an axis to represent the latent state at a certain timestep. Hence, the architecture of the observation module is as follows:

**Decoder**

| Type | Size/Ch. | Act. Fn. | Comments |
|---|---|---|---|
| Latents | $d_\sigma + d_\delta + R_d$ | | |
| Broadcast | $d_\sigma + d_\delta + R_d + 2$ | | +coordinates |
| Conv $5 \times 5$ | 32 | ELU | |
| Conv $5 \times 5$ | 32 | ELU | |
| Conv $5 \times 5$ | 32 | ELU | |
| Conv $5 \times 5$ | 32 | ELU | |
| Conv $5 \times 5$ | 4 | Linear | RGB+Mask |

We follow implementation by OP3 to model object interactions using MLP as follows:

**Dynamics Network**

| Function | Size/Ch. | Act. Fn. |
|---|---|---|
| $d_o$ | 128 | ELU |
| $d_a$ | 32 | ELU |
| $d_{\text{ao-eff}}$ | 128 | ELU |
| $d_{\text{ao-act}}$ | 128 | Sigmoid |
| $d_{\text{oo-eff}}$ | 256 | ELU |
| $d_{\text{ao-act}}$ | 256 | Sigmoid |
| $d_{comb}$ | 256 | ELU |
| $f_{det}$ | 128 | |
| $f_{sto}$ | 128 | |

The architecture is a modification of the interaction function proposed by Van Steenkiste et al. (2018) by conditioning on actions. Dynamics is composed of effects of the action and effects from other objects, modeled as a summation of pairwise interactions. Overall, it can be expressed as,

$$\tilde{\delta}_k = d_o(z_k^{(t)}) \quad \tilde{a} = d_a(a^{(t)}) \quad \tilde{\delta}_k^{act} = d_{ao}(\tilde{\delta}_k, \tilde{a})$$

$$\delta_k^{interact} = \sum_{i \neq k}^{K} d_{oo}(\tilde{\delta}_i^{act}, \tilde{\delta}_k^{act}) \quad z_k^{(t+1)} = d_{comb}(\delta_k^{act}, \delta_k^{interact}), \tag{6}$$

where $d_{ao}$ and $d_{oo}$ are modeled using two neural networks to predict how and to what degree other actions or objects affect an object as, $d_{ao}(\tilde{\delta}_k, \tilde{a}) = d_{\text{ao-eff}}(\tilde{\delta}_k, \tilde{a}) \cdot d_{\text{ao-att}}(\tilde{\delta}_k, \tilde{a})$ and $d_{oo}(\tilde{\delta}_i^{act}, \tilde{\delta}_k^{act}) = d_{\text{oo-eff}}(\tilde{\delta}_i^{act}, \tilde{\delta}_k^{act}) \cdot d_{\text{oo-att}}(\tilde{\delta}_i^{act}, \tilde{\delta}_k^{act})$, respectively. $z_k^{(t)}$ is the concatenation of the deterministic component and stochastic component (relational features). The output, $z_k^{(t+1)}$, is split into deterministic component and relational features using separate networks, $f_{det}$ and $f_{sto}$, respectively.

## C    TRAINING AND EXPERIMENT SETUP

For both datasets, we used Adam optimizer and a batch size of 80. We used gradient clipping where if the norm of global gradient exceeds 5.0, then the gradient is scaled down to that norm. As OP3 (Veerapaneni et al., 2020), for better convergence, we divided the latent distribution into a stochastic component (size $R_s$) and deterministic component (size $R_d$), and set the dimensions for both to 64. Dimensions of global and relational features were both set to 64. We implemented a curriculum training scheme that increases prediction horizon throughout training.

### C.1    BLOCKS DATASET

The 3D block moving dataset, proposed by Janner et al. (2019), is composed from 10000 sequences of moving around two different-colored objects. Each sequence consist of not only images but also action vectors that represent the picking and dropping coordinates. The actions are biased such that 30% of actions will pick up a block and place it in a random position, 40% of actions will pick up a block and place it on top of another block, and 30% of actions are try to pick from a random place and place it in another random position. The dataset also has a downstream planning task to evaluate the quality (how "good") of the representations that the model has learned is. The task consists of a start image (or sequence) and goal image, and we use the model to plan the actions necessary to reach the goal state. The task is split by the number of goal objects, $n = [1, 2, 3]$. For a goal with $n$ objects, we plan and execute $n$ steps in total as a single step is sufficient to move each object to its goal configuration. We repeat the planning and executing for $2n$ times or until it has reached the goal state. Accuracy is computed as $\frac{\text{\# of blocks in the correct position}}{\text{\# goal blocks}}$, where a correct position is based on a threshold value of the distance error.

**Training.** We trained the model with a fixed learning rate of 0.0005 for 400 epochs. The coefficients of KL divergence terms in the loss function were set to 0.01 for relational feature and 0.1 for global feature. Iterative inference was conducted with 4 refinement steps. The number of slots were set to 4.

**Latent Swapping Task.** In order to evaluate whether the model has learned to disentangle global and relational features in a meaningful manner, we swapped either variables between sequences and reconstructed them. More precisely, for any two sequences in the test set, we first inferred their latent variables. Using the mask size of the reconstructed images of each slot, we can identify whether a slot includes the object representations. As we assume that a certain object is tracked in the same slot through time, we used the first timestep to identify the slots with objects and use the same indices for all timesteps in the sequence. Then, we swapped either the global or the relational features of the slots that include object representations between the sequences. The features of the remaining two slots were kept unswapped. Finally, we reconstructed the images using the swapped features.

Throughout the sequences, objects' color remain unchanged while objects' position were affected by the actions. Therefore, the model should have learned to disentangle colors as global features and positions as relational features. In order to evaluate this quantitatively, we used MSE of average RGB of the slots and IoU of slots' masks for global and relational features, respectively. Note that the slots without object representations were not used because their representations were not swapped.

**Planning Task.** As the model has learned object-centric representations, it can be used to identify the pick location. In order to determine the position of a certain object, $k$, we sampled coordinates from a uniform distribution over the $(x, y)$ space and then calculated the weighted average, using its attention coefficient $p(\delta_k | x, y) = \sum_{i \neq k}^{K} d_{\text{oo-att}}(\tilde{\delta}_i^{act}, \tilde{\delta}_k^{act})$ from the dynamics module as its weight. For each action, we sampled actions 5 times using the CEM method. Given representations of

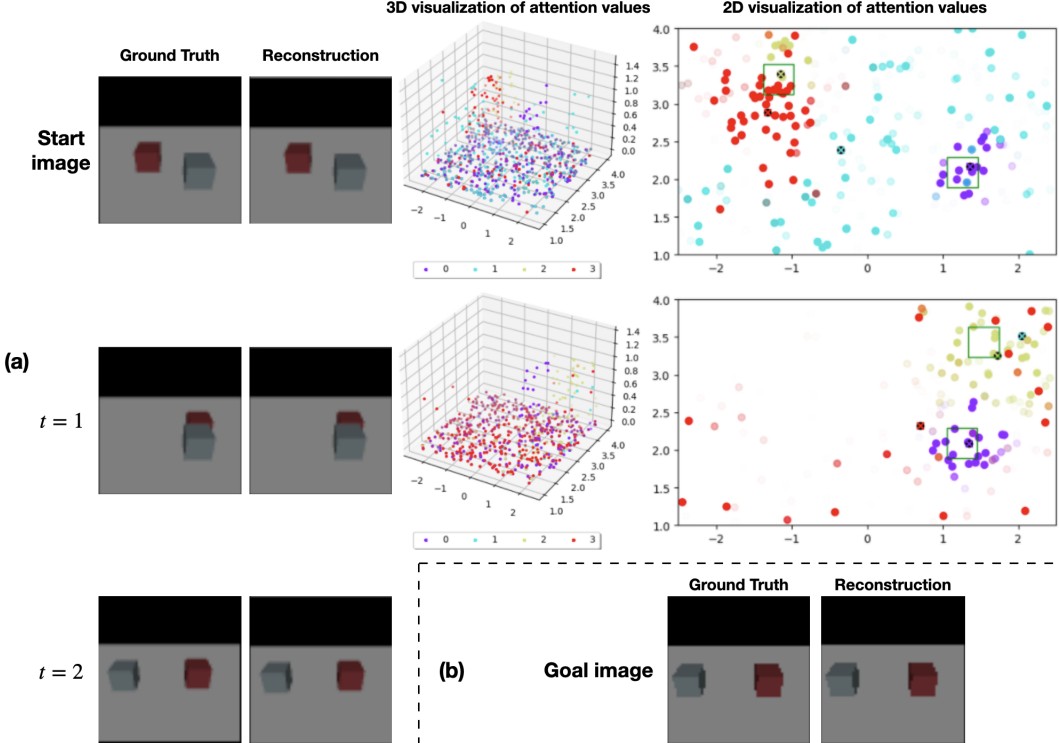

Figure 5: An example of planning results on the Blocks dataset. (a) Each row corresponds to each timestep during planning ($t = 0, 1, 2$). The images in each column show, from left to right, ground truth image, reconstructed image, three-dimensional visualization of attention values where each color corresponds to each slot, and two-dimensional visualization of attention values. For the visualization of attention values, the $x, y$ axes match the environment's coordinates. Points were randomly sampled. Ground truth position of the blocks are shown in green squares in the two-dimensional visualization. Black dots with color crosses show the actual pick location for each slot. Results show that attention values change corresponding to changes in objects' positions. In general, attention values are higher where the object is and smaller in other locations. We also see that objects are tracked in the same slots, yellow and purple. (b) Ground truth and reconstructed images of the goal configuration. Compared with the images on the left ($t = 2$), we see that STEDIE correctly reached the goal configuration.

current timestep, $h_{1:K}^{(t)} = Concat(\sigma_{1:K}, \delta_{1:K}^{(t)})$ and inferred goal representations, $h_{1:K}^g$, we used a cost function, $\mathcal{C}(h_{1:K}^{(t)}, h_{1:K}^g) = \sum_{k \in \mathcal{S}^{(t)}} \min_{l \in \mathcal{S}^g} c(h_k^{(t)}, h_l^g)$ where $\mathcal{S}^{(t)}$ and $\mathcal{S}^g$ denote predicted entities of current timestep and goal image, respectively, to calculate how "good" a certain action is. $c(\cdot)$ can be any distance function, but we found that reconstruction MSE results in the best performance of both OP3 and STEDIE.

## D  VISUALIZATION OF PLANNING

In the planning task, we calculated attention coefficients, $p(\delta_k | x, y) = \sum_{i \neq k}^{K} d_{\text{oo-att}}(\tilde{\delta}_i^{act}, \tilde{\delta}_k^{act})$, to find the pick location. Figure 5 shows one example of the planning result and how the location of attention changes in alignment with the objects' positions.

### D.1  WEIGHTS DATASET

The 2D weighted-block pushing dataset, proposed by Ke et al. (2021), consists of 2D scenes with multiple, different-colored objects. The colors of the objects represent their "weights", such that only a heavier object can move a lighter object while pushing, but vice versa will result in both

objects not moving at all. The dataset consists of three variations, namely, Observed, FixedUnobserved, and Unobserved datasets. In the Observed dataset, all objects have the same shade of color and its intensity represents weight, such that darker-colored objects can move lighter-colored objects, but not vice versa. In the other datasets, objects' colors are sampled from a discrete colormap. For FixedUnobserved dataset, the shapes are coupled with colors and can be interpreted as a hint to identifying its weight. Meanwhile, in the Unobserved dataset, the shapes are distractors and the color defines objects' weights.

**Training.** We trained the model with a fixed learning rate of 0.00067 for 1000 epochs. The coefficients of relational feature and global feature's KL divergence terms in the loss function were set to 0 initially and linearly increased to 0.01 and 0.1, respectively, after the 500th epoch in a range of 100 epochs. Iterative inference was conducted with 5 refinement steps. Number of slots were set to 4. To train OP3, we conducted a grid search on the hyperparameters with batch size of $[16, 32, 80]$, learning rate of $[0.0001, 0.0003]$, and KL coefficient of $[0.001, 0.01]$. From the search, we found that batch size of 16, learning rate of 0.0003, and KL term's coefficient of 0.001 resulted in the best results.

**Causal Understanding Task.** To evaluate if the trained model has learned the causal relationship of objects, we generated a random sequence and visualized the rollout results for all possible actions. At any timestep, the agent could choose to move a certain object one grid horizontally or vertically, or move nothing.

We saw that the attention values were higher around the place where the objects were positioned and lower otherwise. In this example, the slots shown in yellow and purple correspond to the red and gray cubes, respectively. The results also showed that STEDIE was able to track objects in the same slot, which was crucial to solve the planning task because the model needed to compare for each object whether it had reached the goal position or not.

## E    ROLLOUT ANALYSIS

Although OP3 motivates the model to keep track of objects using interactive inference, we found that it fails to keep track of temporally invarying object features (e.g. color). This is because it used the full representation to predict object interactions; object features that are unaffected by interactions were processed through the neural network, and caused the accumulation of small variances. We also found that this caused the object to move between slots across time. These problems have a large impact on downstream task performance as the model cannot identify the "same" object between the current timestep and the goal state. Figure 6 and Figure 7 show examples of comparison between OP3 and STEDIE in terms of rollout, and Figure 8 shows MSE of mean color of slots, which include the objects.

We saw that STEDIE was more capable of tracking objects better as it learned to disentangle interaction-irrelevant features as global features and use only relational features to predict interactions. On the other hand, we observed that OP3 lacks both spatial and temporal consistency.

## F    ABLATIONS

We provide ablation results on interactive inference to infer global features and relational features in Table 5. Following OP3, we updated the parameters at each timestep to ensure timewise consistency for both features. Although the model can learn to reconstruct the video without interactive inference, this leads to poor planning performance as object identity is swapped between slots across time.

## G    COMPARISON WITH OTHER RELATED WORKS

In this section, we provide a more detailed comparison with related works on object-centric learning for multi-object videos (Table 6).

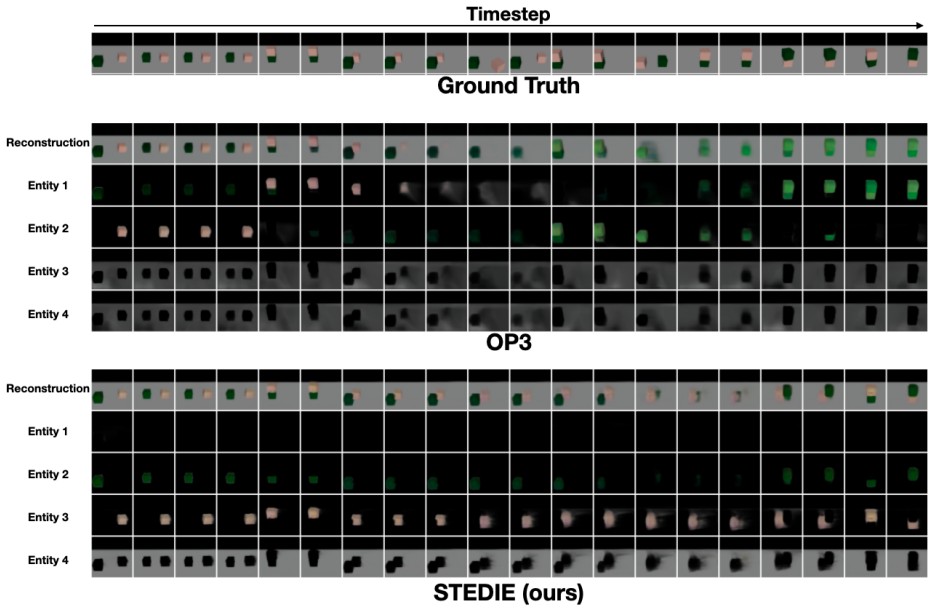

Figure 6: Long-term rollout results of OP3 and STEDIE on the Blocks dataset. Columns represent timestep. Both models were trained with sequences with a maximum length of 5. Although STEDIE does not reconstruct all timesteps accurately, it is able to track the same object.

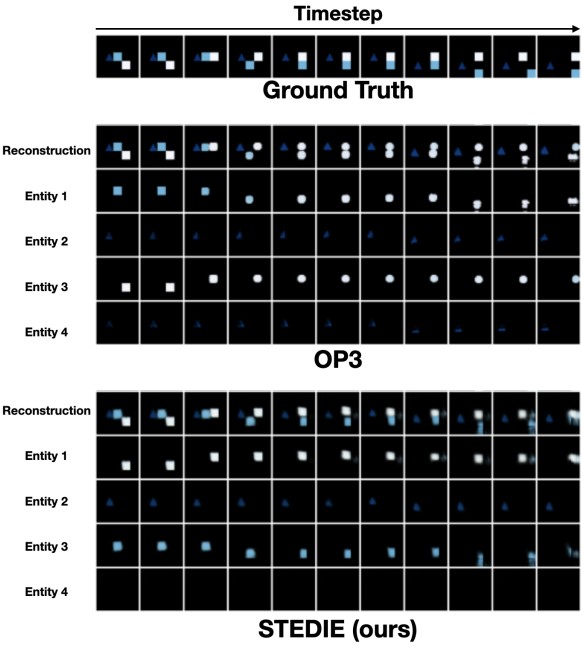

Figure 7: Long-term rollout results of OP3 and STEDIE on the Weights dataset. Columns represent timestep. Both models were trained with sequences with a maximum length of 5. Similar to the rollout example on the Blocks dataset, STEDIE is able to track the same object, leading to better causal understanding.

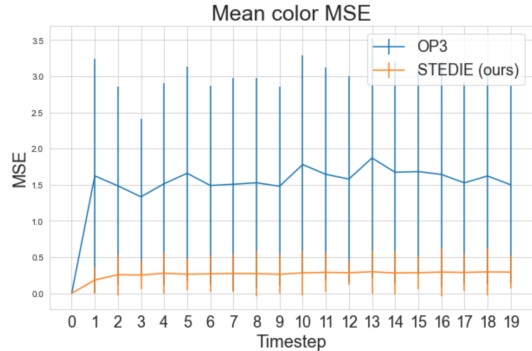

Figure 8: MSE of mean color for slots which include the objects in the Blocks dataset. Mask sizes of the slots were used to identify whether a slot includes information about an object or not. STEDIE is able to reduce color MSE by a large margin, as it ensures temporal consistency by disentangling based on interactions.

Table 5: Ablation results on multi-step planning for building block towers on non-/usage of interactive inference (int. inf.). Evaluation metrics are accuracy (%), MSE, and average CEM steps until reaching goal state. N/A indicates that the model could only predict not to move objects already at their goal position.

| # Blocks | STEDIE | Acc. (↑) | MSE (↓) | Average CEM steps (↓) |
|---|---|---|---|---|
| 1 | w/o int. inf. | 26% | 0.0035 | 1.00 |
|   | w/ int. inf. | **84%** | **0.0012** | **1.00** |
| 2 | w/o int. inf. | 15% | 0.0071 | **1.00** |
|   | w/ int. inf. | **63%** | **0.0041** | 1.45 |
| 3 | w/o int. inf. | 19% | 0.0141 | N/A |
|   | w/ int. inf. | **55%** | **0.0075** | **1.82** |

Table 6: Detailed comparison of relevant prior works on object-centric and its utilization for model-based RL. The models can be categorized based on whether it models interactions, is action-conditioned or not, type of factorization of object representations, and objective function.
†: PARTS (Zoran et al., 2021) expands OP3 by using transformer-based dynamics module instead of modeling pair-wise interactions and replacing the prior as Gaussians that are independent of the timestep, past latents, and actions. In order for the global and relational features to capture interaction-relevant and interaction-irrelevant features, respectively, we think it is crucial that the generative process is conditioned on the actions. Our experimental results show that this also relates with solving OP3's limitations which were raised in PARTS, that "interactions may cause the latent variable to effectively lose its slotted structure and slot independence as time passes in the sequence". We show in our experiments that STEDIE disentangles object representations by conditioning on actions, and that it leads to better reconstruction, object tracking, and scene understanding.

| Method | Models Interactions | Action-Conditioned (Plannable) | Type of factorization | Objective fn. |
|---|---|---|---|---|
| ViMON | ✗ | ✗ | single | ELBO |
| TBA | ✗ | ✗ | single | Pixel-based |
| DDPAE | ✗ | ✗ | implicitly factorized | ELBO |
| SQAIR, SCALOR | ✗ | ✗ | explicitly factorized | ELBO |
| CSWM, SAVI, SAVI++ | ✔ | ✗ | single | Pixel-based |
| OP3 | ✔ | ✔ | single | ELBO |
| PARTS | ✔ | ✔† | single | ELBO |
| GSWM | ✔ | ✗ | explicitly factorized | ELBO |
| STOVE | ✔ | ✔ | explicitly factorized | ELBO |
| STEDIE (ours) | ✔ | ✔ | implicitly factorized | ELBO |

