# OpenReview forum: "Interaction-Based Disentanglement of Entities for Object-Centric World Models"
_ICLR.cc/2023/Conference — ICLR 2023 poster_

### Official Review · Reviewer_sHVk · 2022-10-21

**Confidence:** 4
**Correctness:** 3
**Technical Novelty And Significance:** 3
**Empirical Novelty And Significance:** 3
**Recommendation:** 6

**Clarity, Quality, Novelty And Reproducibility:**

There are som typos:
•	(a) (a) (b) in the caption of Figure 1 should be (a) (b) (c).
•	Does a^{(t)} should be a^{(t-1)} in the first row of page 5?

**Strength And Weaknesses:**

Strength:
The paper presents a very interesting idea of disentangling representations based on interaction.

Experiments show that the learned representations achieve better performance in planning tasks and understanding causal relationships.

Weakness:
Missing discussion on type of factorization, especially between implicitly factorized representation and explicitly factorized representation.

Please discuss why the performance of OP3 on the Weights dataset is so poor even when doing nothing, because the result on the block moving dataset seems fine.

I would like to see the video prediction performance comparison between OP3 and STEDIE on the non-action-conditioned datasets, like CLEVRER.



**Summary Of The Paper:**

This paper proposes a structured world model which learns to disentangle image sequences into interaction-irrelevant global features and interaction-relevant relational features without supervision. The design of the model continues the ideas from IODINE, which iteratively inference latent representations to generative target images. Differently, the global features and relational features are designed to be separated and learned implicitly. The model is evaluated on two downstream tasks: planning and casual understanding, experiments show the effectiveness of the disentangled representation.

**Summary Of The Review:**

The paper presents a method to disentangle representations based on interaction. The ideas are validated quantitatively and qualitatively. Yet, I am concerned about the insufficient evaluation. I am open to adjusting my rating if the authors do sufficiently address the points raised in weaknesses.

---

> ### Author Response · Authors · 2022-11-18
> **Response**
>
> Thank you for your valuable feedbacks. We are pleased to hear that you find our paper interesting. We have added new materials and revised our paper to address the weaknesses you pointed out.
>
> >Missing discussion on type of factorization, especially between implicitly factorized representation and explicitly factorized representation.
>
> Thank you for bringing this up. We have added this discussion in the revised paper (section 4.2, 5). To summarize, there are mainly two differences between STEDIE and STOVE:
>
> (1) STOVE factorizes object representations explicitly by assigning part of the representations as features that represent position, size, and velocity while STEDIE learns to factorize global and relational features unsupervised.
>
> (2) The encoder of STOVE attends the image recurrently for each object, similar to AIR ([Eslami, et al. 2016]), while STEDIE is uses IODINE, a spatial-mixture model, as the encoder.
>
> Although STOVE’s original paper shows its capability to model object interactions in several datasets, as it only infers objects position, velocity, and size from the images, it does not take object color into account to predict the interactions. Altering the factorization to explicitly model object color would possibly make STOVE predict well for the Weights dataset, but designing explicit factorization for each type of dataset is, in general, infeasible.
>
> We would also like to point out that reconstruction MSE of STOVE was not poor, showing 115.86 and 122.31 at t=5,10 respectively for Observed dataset. This shows that STOVE’s explicit factorization was inadequate for the Weights dataset, as it was not able to extract the information needed, object colors, to model the dynamics.
>
> >Please discuss why the performance of OP3 on the Weights dataset is so poor even when doing nothing, because the result on the block moving dataset seems fine.
>
> Thank you for pointing this out. We reconducted a thorough search on training OP3 on the Weights dataset, and was able to improve the model (see Table 4 and Figure 4). The range of grid search is described in the Appendix (section Appendix C.2). Although the results in Figure 4 shows competitive performance of OP3 against STOVE, Table 4 reveals that OP3’s performance drops by a large margin when rolling out predictions further into the future. This suggests that without factorizing object representations, it is not able to capture the causal relationship in the dataset accurately.
>
> >I would like to see the video prediction performance comparison between OP3 and STEDIE on the non-action-conditioned datasets, like CLEVRER.
>
> Thank you for your suggestion. As our main interest was utilizing our model for model-based RL to solve tasks such as planning, we conducted our experiments on datasets with discrete action vectors. We reason that models will struggle to understand object interactions without action inputs as the problem will become more open-ended. For example, in the Weights dataset, pushing a lightly-colored object towards a darker-colored object will result in no change in object positions as the latter object is heavier than the former. However, the same outcome can be observed by trying to push a position where there is no object is present or applying no actions at all. Action information that the agent tried to push the lightly-colored object towards the darker-colored object is crucial in understanding the causal relationship.
>
> Therefore, we would like to leave experiments on non-action-conditioned datasets for future work.
>
> >There are som typos: • (a) (a) (b) in the caption of Figure 1 should be (a) (b) (c). • Does a^{(t)} should be a^{(t-1)} in the first row of page 5?
>
> Thank you for indicating the typos in detail. We’ve proof-read our paper and fixed the typos that were present in the paper. As for the second typo you mentioned, this equation was taken directly from the original OP3 paper (section 4.2 of their paper).

---

> > ### Comment · Reviewer_sHVk · 2022-11-25
> > **Comments after review**
> >
> > I would like to thank the authors for the rebuttal. At this time, I think the contribution is clear and well-proven: disentangling objects' interaction-relevant and interaction-irrelevant features to improve long-term reconstruction quality after interactions with actions and improve the planning and causal understanding performance.
> >
> > However, the experiments are still not solid to prove the significance of this model. Considering that the main contribution is feature disentanglement, I think more analysis could be done in the representation level comparing with other related models, e.g., GSWM, which explicitly factorizes position and contexts, and shows better long-term prediction performance than OP3. What if actions are considered based on GSWM to conduct planning tasks?

---

> > > ### Author Response · Authors · 2022-11-29
> > > **Reply to the comments**
> > >
> > > Thank you for your reply. We are glad to hear that the rebuttal clarified your view on the paper’s contribution.
> > >
> > > > Considering that the main contribution is feature disentanglement, I think more analysis could be done in the representation level comparing with other related models, e.g., GSWM, which explicitly factorizes position and contexts, and shows better long-term prediction performance than OP3.
> > >
> > > Although we could not prepare a qualitative comparison between our proposed model and STOVE in the Weights dataset due to STOVE’s implementation, which requires at least 2 timesteps to infer the entity’s velocity, we believe that the results we reported are sufficient to show the significance of implicit factorization of entities.
> > >
> > > Firstly, the latent swapping experiment (section 4.1) reveals how the global and relational variables capture different aspects of the objects. The results show that our model achieves factorization of objects without explicitly defining what kind of information should be learned as each of the variable. Such implicit factorization in complex, interactive environments is, to the best of our knowledge, has not been explored. Secondly, in the experiments using the Weights dataset (section 4.2), we compare our model against prior works on object-centric learning, including explicit factorization method, STOVE. As Table 4 shows, STOVE was unable to capture the causal relationship as its rollout MSE was poorer than its reconstruction MSE, which showed an MSE as low as STEDIE even when trajecting longer into the future.
> > >
> > > Overall, we think the experimental results show (1) the **limitation of explicit factorization** in which a small change in dataset setup requires changes in factorizing, and (2) the **significance of implicit factorization** as it learns to disentangle the scene into objects and objects into global and relational features.
> > >
> > > > What if actions are considered based on GSWM to conduct planning tasks?
> > >
> > > Thank you for your suggestion. As we wrote in the conclusion, we think that the comparison between explicit factorization and implicit factorization should be explored in future works as well, because there are architectural differences between models as well. We agree that extending GSWM for environments with actions by an agent is one approach to be considered. However, we did not investigate this option in this paper as extending GSWM to create a new plannable, explicit factorization model was not trivial. Although the approach would be to condition the propagation module of GSWM on actions, this requires a thorough investigation. As GSWM’s propagation module is composed of not only a single RNN but multiple networks that model interactions with other objects and background in parallel, how to take the action information into regard is unclear. Therefore, we believe that this is outside of the scope of this paper, and wish to leave this question open for future works in this area of research.

---

> > > > ### Comment · Reviewer_sHVk · 2022-12-02
> > > > **Reponse to the Reply**
> > > >
> > > > Thanks for the author's reply. Although extending GSWM is valuable, it needs too much work in this short discussion period. I am more positive on this paper and would update my rating to 7 (between marginally above and accept).

---

### Official Review · Reviewer_Hvkp · 2022-10-22

**Confidence:** 2
**Correctness:** 3
**Technical Novelty And Significance:** 3
**Empirical Novelty And Significance:** 3
**Recommendation:** 6

**Clarity, Quality, Novelty And Reproducibility:**

Pretty clear, novel and seems reproducible. However, i think the downstream task's quality is a bit lacking.

**Strength And Weaknesses:**

I want to preface this by saying that I'm not really familiar with the field of object-centric learning.

Strength:
- I think the model is well designed and has some interesting novelties. The authors do a good job explaining the model.

Weakness:
- Equation one \sigma has a (0) superscript, which seems to be a typo.
- I think overall, the downstream task evaluation seems a bit underwhelming to me.
1. I think the evaluation metrics and visualizations are not exactly informative. As far as I can tell from the paper, the action and measurements are done in 3D. The evaluation metrics of accuracy is based off a distance metric that's unspecified in the paper and appendix. Not showing direct measurement in 3D limits my confidence in the model. MSE is calcuated in pixel space, which is weird, because say we move the cube along the camera direction, it results in not so much change in pixel mse, but the 3D location can be far off.
2. From what I can tell, OP3 seems like a model from quite awhile ago, while STOVE seems to be a very relevant baseline. I can't really tell from the paper which reference is STOVE, and why the authors are not comparing with it.

**Summary Of The Paper:**

The paper propose STEDIE, a new model that disentangles object representations based on interactions, into interaction-relevant relational features and interaction-irrelevant global features without direct supervision. Across different experiments, the authors showcases the model's effectiveness.

**Summary Of The Review:**

Please see my strength and weakness.

---

> ### Author Response · Authors · 2022-11-18
> **Response**
>
> Thank you for your valuable comments. We are pleased to hear that you find our proposal novel and well-designed. We have added new materials and revised our paper to address the weaknesses you pointed out.
>
> >Equation one \sigma has a (0) superscript, which seems to be a typo.
>
> Thank you for pointing this out. We’ve proof-read our paper and fixed the equation as well as other grammatical errors that were present in the paper.
>
> >I think the evaluation metrics and visualizations are not exactly informative. As far as I can tell from the paper, the action and measurements are done in 3D. The evaluation metrics of accuracy is based off a distance metric that's unspecified in the paper and appendix. Not showing direct measurement in 3D limits my confidence in the model. MSE is calcuated in pixel space, which is weird, because say we move the cube along the camera direction, it results in not so much change in pixel mse, but the 3D location can be far off.
>
> Thank you for bringing this up. As you have mentioned, we agree that accuracy in the RGB space has limitations in capturing the error of models as the Blocks dataset is composed of 3D objects. However, we used this evaluation metric as it was the only metric that was used in the original OP3 paper ([Veerapaneni, et al. 2019]). We would like to leave exploring additional evaluation metrics on this dataset as future work.
>
> >From what I can tell, OP3 seems like a model from quite awhile ago, while STOVE seems to be a very relevant baseline. I can't really tell from the paper which reference is STOVE, and why the authors are not comparing with it.
>
> Thank you for your feedback. Although STOVE is relevant as it learns object representations for multi-object videos unsupervised and can be conditioned on actions, there are mainly two differences:
>
> (1) It factorizes object representations explicitly by assigning part of the representations as features that represent position, size, velocity while STEDIE learns to factorize global and relational features unsupervised.
>
> (2) The encoder of STOVE attends the image recurrently for each object, similar to AIR ([Eslami, et al. 2016]), while STEDIE is uses IODINE, a spatial-mixture model, as the encoder.
>
> STOVE explicitly infers only objects position, velocity, and size from the images x_t and does not infer object color. As color information is crucial in understanding the dynamics of the Weights dataset, STOVE cannot model the interactions as we show in the revised paper (Table 4). Altering the factorization to explicitly model object color would possibly make STOVE predict well for Weights dataset, but designing explicit factorization for each type of dataset is, in general, infeasible.
>
> In the revised paper, we have added comparisons against STOVE for Weights dataset. Experimental results show that as STOVE explicitly factorizes objects’ position, velocity, and size only, it struggles to understand the causal relationships of the objects in the dataset. We did not experiment on the Blocks dataset as STOVE uses AIR-based encoder as its architecture, which is known to struggle with datasets with background information [Lin, et al. 2020]. Finally, we would like to note that architectural difference STOVE and OP3/STEDIE may had an effect to the difference in their performances. We would like to leave implementing explicit factorization using spatial-mixture models as future work.

---

### Official Review · Reviewer_qwR6 · 2022-10-24

**Confidence:** 3
**Correctness:** 3
**Technical Novelty And Significance:** 3
**Empirical Novelty And Significance:** 3
**Recommendation:** 5

**Clarity, Quality, Novelty And Reproducibility:**

Clarity and quality: Although not an expert, I find the paper writing and logic flow clear and easy to follow. The comparison with prior work on object-centric methods is also clear and distinctive.
Typo: section 3.2 dynamic module

Novelty: The proposed model is built upon previous work OP3 and various strategies are applied to infer the parameters. The major novelty lies in the disengagement probabilistic design.

Reproducibility: the paper included implementation details in the supplementary material. Code is not included.


**Strength And Weaknesses:**

Strengths:
- This paper differentiates itself from prior work by disentangling the object-centric representation to interaction-relevant feature and the global feature, which is a promising direction for unsupervised representation learning.
- The experiments show that the model can learn disentangled representations, and the learned representations can facilitate downstream tasks in planning and roll-out reconstruction.

Weaknesses:
The major concern lies in the experiments. 1) the latent swapping experiments are conducted on the block moving datasets, and the involved features are relatively simple, i.e., color for global feature and position as relational feature. Prior work also demonstrates such disentangled representation even without explicit modeling of the dynamic and static features. Without further quantitative comparison with other baselines or in more complex scenarios, it’s hard to justify the significance of the proposed model. 2) The downstream experiments are mainly compared against OP3, which does not disambiguate global or relational features. Are there justifications why not compare to factorized models like STOVE?


**Summary Of The Paper:**

This paper proposes a probabilistic model to learn disentangled object representation in an unsupervised fashion. The generative model is expressed through an observation module and a dynamic module, where the representations of entities are factorized into interaction-relevant relational features and interaction-irrelevant global features. The model is optimized with variational inference by maximizing the ELBO. Latent swapping experiments and downstream tasks are conducted on Block and Weights datasets.

**Summary Of The Review:**

My ratings are given based on my evaluation of the strengths and weaknesses above. The authors are expected to address my concerns over the experiments.

---

> ### Author Response · Authors · 2022-11-18
> **Response**
>
> Thank you for your thorough feedbacks. We are pleased to hear that you find our proposed method promising and novel. We have added new materials and revised our paper to address the weaknesses you pointed out.
>
> >The latent swapping experiments are conducted on the block moving datasets, and the involved features are relatively simple, i.e., color for global feature and position as relational feature. Prior work also demonstrates such disentangled representation even without explicit modeling of the dynamic and static features. Without further quantitative comparison with other baselines or in more complex scenarios, it’s hard to justify the significance of the proposed model.
>
> Thank you for your comment. We would like to clarify how our proposed model differs from relevant prior works, namely OP3 and STOVE, below:
>
> - Compared to OP3, we introduce a novel implicit factorization of object representations, into global and relational features. By factorizing, results (Table 2 and Figure 3) show that factorization leads to better performance in downstream tasks. As shown in Figure 6-8, this is because global features work as a key to identify the “same” object in the sequence. Whereas OP3 learned to take a shortcut by modeling the movement of an object by using multiple slots to model each as separate objects, STEDIE forces the model to track the object in the same slot. As a result, STEDIE is able to learn the dynamics of object interactions more accurately.
> - Compared to STOVE, we model to factorize implicitly, whereas STOVE modeled the factorization explicitly to infer object positions, sizes, and velocities from images. The results on the Weights dataset (Table 4) shows how explicit factorization needs to be adjusted for each kind of dataset. For the Weights dataset, color information is needed to model object dynamics accurately, which is not modeled for STOVE. On the other hand, since STEDIE models the factorization implicitly, it is able to adjust what information is learned in each feature depending on the dataset.
>
> Action-conditioned implicit factorization of object representations is, to the best of our knowledge, has not been explored in prior works and we think that it is a type of factorization that can be applied for various types of datasets and is useful for solving downstream tasks.
>
> >The downstream experiments are mainly compared against OP3, which does not disambiguate global or relational features. Are there justifications why not compare to factorized models like STOVE?
>
> Thank you for pointing this out. We have added comparisons against STOVE for the Weights dataset. Experimental results show that as STOVE explicitly factorizes objects’ position, velocity, and size only, it struggles to understand the causal relationships of the objects in the dataset. Altering STOVE’s factorization to explicitly model object color would possibly make it predict well for the Weights dataset, but designing explicit factorization for each type of dataset is, in general, infeasible. We did not experiment on the Blocks dataset as STOVE uses AIR-based encoder as its architecture, which is known to struggle with datasets with background information [Lin, et al. 2020]. Finally, we would like to note that architectural difference STOVE and OP3/STEDIE may had an effect to the difference in their performances. We would like to leave implementing explicit factorization using spatial-mixture models as future work.

---

> > ### Comment · Reviewer_qwR6 · 2022-12-13
> > **Post rebuttal**
> >
> > After carefully reading the rebuttal and the reviews from other reviewers, I'm inclined to hold my original opinion as borderline. I appreciate the extra experiments the author provided, especially the comparisons against STOVE. However, as also mentioned by other reviewers, the current experiment results and figures are not strong enough to prove the significance of this model, i.e., the implicit factorization of object representations. More evidence is expected on the benefits of implicit factorization against explicit factorization in 1) the ability to model more complex features other than color and position and 2) the validity when explicit factorization fails. That being said, I'm more confident about the paper after the rebuttal and think it would be a good submission if the common concerns are addressed.

---

### Official Review · Reviewer_wuf5 · 2022-10-26

**Confidence:** 4
**Correctness:** 3
**Technical Novelty And Significance:** 4
**Empirical Novelty And Significance:** 4
**Recommendation:** 6

**Clarity, Quality, Novelty And Reproducibility:**

The paper is fairly clear, but the authors should improve the writing and how they describe things I mentioned in the weaknesses section. The results seem of quality to make the point the paper is about. The method seems novel, and the authors provided details of the architecture in the supplementary material so it should be reproducible.

**Strength And Weaknesses:**

Strengths:
+ New method for disentangled object centric representation learning
+ Experiments showing the disentanglement is present
+ Experiments showing the disentanglement leads to better downstream performance


Weaknesses:

- Intersection over union evaluation:
The authors present an intersection over union evaluation to show that interaction-relevant latents contain location information. However, I am not clear how the authors are able to get locations from the latents, and so, can't really see how this evaluation is possible. Can the authors clarify how this is done? Or point out where in the paper this is described. If this is in fact somewhere in the paper, and I cannot find it, it would be useful if this is revised later on so it's easier to find.



- Pairwise interactions mechanism visualizations:
In page 5, the authors mention that they use a d_{o,o} function to calculate pairwise interactions between objects. However, there is no analysis of how these interactions are happening after a model is trained. If this is possible, it would be good to have some type of visualization showing how these become active when objects are interacting in the pixel space.



- Typos in the writing:
There are many typos throughout the text. For example, in section 3.2, first paragraph, line 7, module is misspelled as moduele. In addition, there is misuse of the verb is. For example, in page 6, 4th paragraph, the authors say "Finally, parameters of a generative model pθ and an inference model qφ is" while it should be "Finally, parameters of a generative model pθ and an inference model qφ are". These type of typos are happening in multiple places in the manuscript. I suggest the authors proof read the manuscript and fix these writing issues.

**Summary Of The Paper:**

This paper proposes a method that learns disentangled representations of objects determined by interaction-relevant and interaction-irrelevant latents. The proposed method is based on OP3, but with a formulation that encourages the desired disentanglement. In experiments, the authors show that the interaction-relevant and interaction-irrelevant latent variables are in fact disentangled by swapping them between two different sequences. They also show that the disentangled representations lead to improved performance in downstream tasks against the baselines.

**Summary Of The Review:**

In conclusion, I find this paper interesting regardless of the toy setup the conceptual contribution is shown. I have a couple of questions that I have stated in the weakness section that I would like the authors to clarify. In the mean time, I am leaning towards accepting this paper, but I am looking forward to discussions to determine the final assessment.

---

> ### Author Response · Authors · 2022-11-18
> **Response**
>
> Thank you for your detailed and thoughtful feedbacks. We are delighted to hear that you find our proposed method and experimental results interesting. We have added new materials and revised our paper to address the weaknesses you pointed out.
>
> >Intersection over union evaluation: The authors present an intersection over union evaluation to show that interaction-relevant latents contain location information. However, I am not clear how the authors are able to get locations from the latents, and so, can't really see how this evaluation is possible. Can the authors clarify how this is done? Or point out where in the paper this is described. If this is in fact somewhere in the paper, and I cannot find it, it would be useful if this is revised later on so it's easier to find.
>
> Thank you for pointing this out. We’ve added the details on the setup and evaluation methods on the latent swapping experiment in the main text (section 4.1) as well as in the Appendix (section Appendix C.1). As we use a BroadcastCNN as our decoder, each entity representations is decoded into pixel-wise mean and segmentation mask. Interaction over union was calculated using the segmentation masks of before and after swapping either global or relational features.
>
> >Pairwise interactions mechanism visualizations: In page 5, the authors mention that they use a d_{o,o} function to calculate pairwise interactions between objects. However, there is no analysis of how these interactions are happening after a model is trained. If this is possible, it would be good to have some type of visualization showing how these become active when objects are interacting in the pixel space.
>
> Thank you for bringing this up. We have added an example of planning results and how attention values change over time in the Appendix (section Appendix D). Figure 5 shows that (1) attention values are large where the objects are located, and (2) objects are tracked in the same slot over time, which lead to high accuracy in planning.
>
> >Typos in the writing: There are many typos throughout the text. For example, in section 3.2, first paragraph, line 7, module is misspelled as moduele. In addition, there is misuse of the verb is. For example, in page 6, 4th paragraph, the authors say "Finally, parameters of a generative model pθ and an inference model qφ is" while it should be "Finally, parameters of a generative model pθ and an inference model qφ are". These type of typos are happening in multiple places in the manuscript. I suggest the authors proof read the manuscript and fix these writing issues.
>
> Thank you for pointing this out. We’ve proof-read our paper and fixed the grammatical errors that were present in the paper.

---

### Comment · Reviewer_wuf5 · 2022-11-18
**No response from authors?**

It looks like authors did not provide a rebuttal or they decided to make their response to other reviewers only visible by those reviewers. Is this the case?

---

> ### Author Response · Authors · 2022-11-18
> **We are preparing for the revised version at the moment**
>
> Thank you for your kind notification. We are currently working on our revised version and the replies to the reviewers. We will release the rebuttal as soon as possible.

---

### Author Response · Authors · 2022-11-18
**Thank you to all reviewers and meta-reviewers!**

We would like to show our sincere gratitude to all reviewers for their detailed and thorough feedbacks. We really appreciate that all reviewers found our proposed model promising and the experimental results interesting.

For this rebuttal, we have added experimental results on training STOVE for the Weights dataset. We have also updated some results on the Weights dataset after adjusting hyperparameters. Furthermore, we have added extra figures in the appendix based on the feedbacks. Finally, we have rechecked our paper for grammatical errors.

---

### Decision · Program_Chairs · 2023-01-20

**Decision:**

Accept: poster

**Justification For Why Not Higher Score:**

This paper is clearly borderline and several concerns around significance of the limited experimental setting (primarily toy 2D and 3D synthetic environments) were raised. Furthermore the quality of presentation could be improved.

**Justification For Why Not Lower Score:**

As mentioned above, the presented idea is both conceptually and technically novel, the experimental evaluation is solid (i.e. claims are verified both qualitatively and quantitatively) but not outstanding, and the claims of the paper are relatively moderate but still significant. Overall, it can be considered for being accepted at ICLR.

**Metareview: Summary, Strengths And Weaknesses:**

This paper addresses the problem of learning a structured, object-decomposed world model of simple synthetic 2D and 3D environments. The novelty lies in how the method disentangles features of objects: it learns to separate object features into interaction-/physics-irrelevant (“global”) properties such as object color and appearance, and dynamic / interaction-relevant properties such as position of an object. This is done via separation of latent variables in the (structured) probabilistic model, based on the OP3 [1] framework. This separation is validated both qualitatively and quantitatively on simple 2D and 3D synthetic environments that involve object-specific actions. The authors further demonstrate quantitative benefits of this separation for planning tasks.

I agree with the reviewers that this paper addresses an interesting and challenging problem (joint action-conditioned video prediction and object decomposition), is fairly well-written and proposes a novel approach to implicitly decompose interaction-relevant and global properties from raw video. The fact that the model can do this without explicitly specifying which features should capture location, size, content etc. of objects is interesting and significant. Concerns are related to the low quality/interpretability of the figures, the limited experimental setup (primarily 2D and 3D toy environments) and the overall technical complexity of the model.

One major concern was that the authors were not comparing against an object-centric baseline with an explicitly factorized latent object representation. In their rebuttal, the authors have added a convincing experimental comparison against STOVE [2] to address this issue. Nonetheless, a comparison against more recent object-centric, action-conditioned video models such as PARTS [3] is missing from the paper and the authors are highly encouraged to at least include a discussion of how their method relates to PARTS [3].

This is very much a borderline paper that would clearly benefit from another round of revisions, but given that the presented idea is both conceptually and technically novel, the experimental evaluation is solid (i.e. claims are verified both qualitatively and quantitatively) but not outstanding, and the claims of the paper are relatively moderate but still significant and of interest for the community, I think it can be considered for acceptance.

As mentioned by the reviewers, the paper would heavily benefit from showing more experimental evidence for disentanglement of global vs. interaction-relevant features beyond just object color and position. Further, the authors are strongly encouraged to improve the quality of the figures (both for model and results) of the paper to make it easier to read and understand.

[1] Veerapaneni et al., Entity Abstraction in Visual Model-Based Reinforcement Learning (CoRL 2019)
[2] Kossen et al., Structured Object-Aware Physics Prediction for Video Modeling and Planning (ICLR 2020)
[3] Zoran et al., PARTS: Unsupervised segmentation with slots, attention and independence maximization (ICCV 2021)

**Note From Pc:**

if the above contains the word "oral" or "spotlight" please see: "oral" presentation means -> notable-top-5% and "spotlight" means -> notable-top-25%. As stated in our emails, we are disassociating presentation type from AC recommendations

**Summary Of Ac-Reviewer Meeting:**

Only reviewers qwR6 and wuf5 were able to attend the meeting, the other reviewers (Hvkp and sHVk) were unable to attend due to time zone constraints and shared their updated opinion after the rebuttal offline (both are in favor of acceptance / borderline acceptance).

Reviewer wuf5 argued for (borderline) acceptance for the following reasons:
* Disentanglement is novel and interesting; seems to be helping the OP3 baseline; interaction-relevant features makes sense
* After rebuttal: Comparisons against STOVE is insightful; the fact that they use implicit disentanglement vs STOVE uses explicit ones is novel

Reasons for rejecting the paper (wuf5):
* Asked authors for evidence how the interaction-relevant vs global features are disentangled
* Authors responded with a difficult to understand plot; some concerns around whether this disentanglement is indeed happening

Reviewer qwR6 initially argued for borderline reject:
* Experimental section: rather complex model,
* Too simple experiment: global features / static features just color
* Previous methods could also potentially do this disentanglement even if not explicit; unclear practical relevance for more complex / realistic environments

Reasons for accepting the paper (qwR6):
* Disentanglement of representation; implicit representation;
* Experiments that demonstrate implicit disentanglement performs better

Reviewer wuf5:
Agrees w/ reviewer qwR6 that experiments are likely too simple; depends on writing of paper

Conclusion: Decision should depend on claims made in the paper – if authors don’t claim too large impact (real-world etc) then this is fine, but would need explicit, strong evidence for actual disentanglement

(comment by AC: real-world impact is not claimed in paper and disentanglement is evident from experiments, although only on simplistic scenes)